# Site-Specific Effects of Online rTMS during a Working Memory Task in Healthy Older Adults

**DOI:** 10.3390/brainsci10050255

**Published:** 2020-04-27

**Authors:** Lysianne Beynel, Simon W. Davis, Courtney A. Crowell, Moritz Dannhauer, Wesley Lim, Hannah Palmer, Susan A. Hilbig, Alexandra Brito, Connor Hile, Bruce Luber, Sarah H. Lisanby, Angel V. Peterchev, Roberto Cabeza, Lawrence G. Appelbaum

**Affiliations:** 1Department of Psychiatry and Behavioral Science, Duke University School of Medicine, 200 Trent Drive, Box 3620 DUMC, Durham, NC 27710, USA; courtney.crowell@duke.edu (C.A.C.); moritz.dannhauer@duke.edu (M.D.); wesley.lim@duke.edu (W.L.); hannah.l.palmer@duke.edu (H.P.); susan.hilbig@duke.edu (S.A.H.); alexb2323@gmail.com (A.B.); connor.hile@duke.edu (C.H.); sarah.lisanby@nih.gov (S.H.L.); angel.peterchev@duke.edu (A.V.P.); greg@duke.edu (L.G.A.); 2Department of Neurology, Duke University School of Medicine, 3116 N Duke Street, Durham, NC 27704, USA; simon.davis@duke.edu; 3Center for Cognitive Neuroscience, Duke University, 308 Research Drive, Durham, NC 27710, USA; cabeza@duke.edu; 4National Institute of Mental Health, 6001 Executive Boulevard, Bethesda, MD 20852, USA; bruce.luber@nih.gov; 5Department of Biomedical Engineering, Duke University, 305 Teer Engineering Building, Box 90271, Durham, NC 27708, USA; 6Department of Electrical and Computer Engineering, Duke University, 305 Teer Engineering Building, Box 90271, Durham, NC 27708, USA; 7Department of Neurosurgery, Duke University School of Medicine, 200 Trent Drive, Box 3807 DUMC, Durham, NC 27710, USA; 8Department of Psychology & Neuroscience, Duke University, 417 Chapel Drive, Durham, NC 27708, USA

**Keywords:** repetitive transcranial magnetic stimulation, working memory, aging, fMRI, electric field modeling

## Abstract

The process of manipulating information within working memory is central to many cognitive functions, but also declines rapidly in old age. Improving this process could markedly enhance the health-span in older adults. The current pre-registered, randomized and placebo-controlled study tested the potential of online repetitive transcranial magnetic stimulation (rTMS) applied at 5 Hz over the left lateral parietal cortex to enhance working memory manipulation in healthy elderly adults. rTMS was applied, while participants performed a delayed-response alphabetization task with two individually titrated levels of difficulty. Coil placement and stimulation amplitude were calculated from fMRI activation maps combined with electric field modeling on an individual-subject basis in order to standardize dosing at the targeted cortical location. Contrary to the a priori hypothesis, active rTMS significantly decreased accuracy relative to sham, and only in the hardest difficulty level. When compared to the results from our previous study, in which rTMS was applied over the left prefrontal cortex, we found equivalent effect sizes but opposite directionality suggesting a site-specific effect of rTMS. These results demonstrate engagement of cortical working memory processing using a novel TMS targeting approach, while also providing prescriptions for future studies seeking to enhance memory through rTMS.

## 1. Introduction

Working memory (WM), the capacity to maintain and manipulate information in a temporary mental buffer, is central to many aspects of human cognition. Indeed, through the interface between long-term memories and moment-to-moment information available in the environment, WM allows humans to organize relevant information in order to carry out successful goal-directed behaviors [1]. As such, WM capacity is intrinsic to many daily activities such as reading, performing arithmetic, and keeping track of ideas during a conversation [2,3]. Both WM capacity and the ability to manipulate content that is held in WM declines with age [4,5]. Therefore, different approaches have been proposed to prevent this decline.

Brain stimulation techniques, such as repetitive transcranial magnetic stimulation (rTMS) and transcranial alternating current stimulation (tACS), have gained increased attention as a means to enhance WM and slow age-related impairments. rTMS uses brief, high-intensity magnetic fields to depolarize neurons underneath a magnetic coil. When applied over a brain region that helps support a specific cognitive function, rTMS has the potential to modulate related behavior. In many such studies, rTMS applied online, i.e., during the performance of a task, has been shown to interfere with ongoing cognitive processes, thus impairing behavioral performance ([6] for a review). In other cases, however, studies have reported performance enhancement when applying online rTMS ([7] for a review). For example, stimulation of the parietal cortex during WM maintenance tasks has resulted in significant decreases in reaction times and improvements in response accuracy [8,9]. These contrasting results suggest that online rTMS may affect performance in a manner that is specific to the ongoing process and the spatio-temporal parameters of stimulation, for example, by modulating endogenous task-related oscillatory dynamics [10]. 

Theta oscillations, in particular, have been shown to play a crucial role in memory processes [11], and studies applying brain stimulation within the theta band (5–8 Hz) tend to demonstrate significant WM performance enhancement, as reported in a recent review investigating the effect of different types of brain stimulation, including tACS and rTMS [12]. For example, using tACS, a neuromodulatory technique that uses oscillatory sinusoidal electrical currents to entrain the brain at a chosen frequency, Violante et al. [13] demonstrated that synchronous theta-tACS (6 Hz) applied over the right fronto-parietal network, while subjects were performing either a simple choice reaction time task, a verbal 1-back task, or a verbal 2-back task, significantly shortened subjects’ reaction times. This effect was found only in the most difficult of the three tasks, with behavioral improvements that correlated with increased BOLD activity in the stimulated network, suggesting that tACS can improve both behavior and alter brain activity in a state-dependent manner. 

The effects of rTMS on WM have also been investigated and have demonstrated positive effects on behavioral performance. Indeed, by applying 5 Hz rTMS over the left intraparietal sulcus during an auditory WM task, combined with electroencephalography recordings, Albouy et al. [14] demonstrated that rTMS significantly increased the amplitude of theta oscillations and that this increase correlated with behavioral performance improvements observed with active stimulation. A similar pattern of results was also found by Li et al. [15], who reported that 5 Hz rTMS applied over the left superior parietal lobe during a verbal WM task significantly improved behavioral performance, and increased theta-oscillations, while also reducing the amplitude of alpha oscillations. Moreover, these oscillatory physiological changes in the theta oscillations were positively correlated with behavioral improvements, indicating a mechanistic link between theta band activity and memory function. 

In addition to periodic stimulation protocols that utilize a single oscillatory frequency, recent protocols have begun to test patterned stimulation protocols that involve multiple, simultaneous oscillations at different frequencies. In particular, intermittent theta-burst stimulation (iTBS), which utilizes intermittent patterns of 50 Hz pulses modulated at 5 Hz, has been shown to induce longer-lasting therapeutic effects for major depressive disorder, with shorter durations of stimulation. Studies that have tested iTBS in the context of WM have produced mixed results, with studies showing both large and significant behavioral improvements [16], as well as small, non-significant effects [17]. Despite the widespread interest in the brain stimulation to enhance WM, multiple methods being tested, and the development of more physiologically informed approaches, overall effect sizes in these studies remain moderate, as reported in a recent meta-analysis [18]. 

Beyond the technical challenges of modulating WM performance with non-invasive brain stimulation, the application of neuromodulation in older adults presents a number of unique challenges. For example, aging has been associated with a relative decrease in the excitability of intracortical inhibitory circuits [19], and is also associated with a decline in cortical thickness [20] and cortical volume [21], pointing towards a need to adjust for these factors when stimulating older adults. While linear adjustments of stimulation amplitude according to the distance from scalp to cortex [22] may provide some correction for the systemic differences between older and younger adult brains, they do not account for the spread of rTMS-induced electric field across the cortical surface. In fact, there is reliable evidence that once multilinear changes in neuroanatomy associated with aging are controlled, the motor-evoked response to TMS does not differ significantly across the lifespan [23], suggesting that electric-field (E-field) modeling is a necessary component to precise dosing of TMS attempting to induce changes in individuals with differing cortical geometry.

Recently, our group tested the effect of 5 Hz rTMS over the left dorso-lateral prefrontal cortex (DLPFC) of healthy young and older adults while participants performed a delayed-response alphabetization task (DRAT), in which they were asked to mentally arrange an array of letters into alphabetical order during a delay period. Results from this study revealed that both younger and older adults showed enhanced accuracy on the DRAT with active rTMS compared to somatosensory-matched electrical sham stimulation. These gains, however, were observed only in the most difficult task conditions, therefore replicating the difficulty-specific effect observed by Violante et al. [13], reported above. Moreover, despite the presence of behavioral gains, the observed effects were small, leading to only a 4% improvement in memory recall [24]. 

As such, the current study aimed to implement a related stimulation protocol in older adults, using a modified targeting approach with the goal of obtaining larger behavioral gains. Given past reports of rTMS-induced performance improvements in WM with stimulation delivered to the parietal cortex (e.g., [9,13,14]), and extant theories that this region is central online attentional processes critical for WM [25,26], this region was targeted in the current study. Moreover, because results from our previous fMRI study indicated that manipulation of information in the DRAT produced the greatest activation in the superior parietal lobule [27], fMRI-BOLD activity in this region was used to derive individualized targets for each participant in the current study. Furthermore, in an attempt to induce approximately the same field strength in the target region across subjects, the current study defined stimulation amplitude according to E-field calculations, rather than resting motor threshold, as is typical of many studies. This computational dosing method accounts for the individual head anatomy and was deployed in an effort to minimize individual variability of the response. 

Based on this experimental design, it was hypothesized that active rTMS would significantly enhance WM manipulation performance and that this effect would be most pronounced in the most difficult task conditions, consistent with the view that cognitive performance is most vulnerable to neuromodulation under the most demanding conditions [9,28,29]. Contrary to this a priori hypothesis, 5 Hz rTMS to the left lateral parietal cortex during the delay period of the DRAT was found to impair WM performance. When considered in light of our former study [24], the effects can be interpreted as being site-specific effects of rTMS on WM manipulation.

## 2. Materials and Methods

### 2.1. Participants

Thirty-nine healthy subjects (60–80 years old) were recruited into this single-blind randomized within-subject controlled trial, which was pre-registered on ClinicalTrials.gov (NCT02767323). All participants provided written informed consent, which was approved by the Duke University Institutional Review Board (#Pro00065334). Participants were excluded if they had any contraindication to TMS or MRI, current or past psychiatric disorders or neurological disease (*n* = 1), or a total scaled score lower than eight on the Dementia Rating Scale-2 [30] (*n* = 1). Participants were also excluded if they tested positive on a urine drug screen (*n* = 2), performed poorly on the WM task during the initial visit (*n* = 11), or experienced non-compliance, for example by responding in the task with random keys presses (*n* = 3) (See Figure 1 for consort diagram). All data were collected in Duke School of Medicine, in the Brain Stimulation Research Center. According to these criteria, 18 participants were excluded, along with another 6 participants who withdrew participation for unspecified reasons. Fifteen subjects completed the full protocol (see Table 1 for baseline demographic). Participants had normal, or corrected-to-normal, vision and were native English speakers. Participants were compensated $20/hour for their efforts with a $100 bonus if they completed all study activities.

### 2.2. Experimental Protocol

Participants were scheduled for 6 sessions: a consenting visit including screening, resting motor threshold assessment, and practice at the delayed-response alphabetization task, followed by an MRI visit and four rTMS visits (Figure 2). The following sections give a brief overview of the methods, and additional information can be found in [24].

### 2.3. Delayed-Response Alphabetization Task (DRAT)

On each trial of the DRAT, an array of letters was presented on the screen for 3 s, followed by a 5-s delay period during which the participants were asked to mentally reorganize the letters into alphabetical order (Figure 3). After the delay period, a letter with a number above it appeared on the screen for 4 s and participants were asked to report via a key press whether the letter was (1) not in the original set, (2) in the original set and the number matched the serial position of the letter once the sequence was alphabetized, or if (3) in the original set but the number did not match the serial position of the letter once alphabetized. These conditions are referred to as ‘new’, ‘valid’, and ‘invalid’, respectively. 

During the first visit, participants performed the DRAT using a staircase procedure to establish individualized difficulty levels. Four individualized difficulty levels were defined according to the intersection between a sigmoid curve, fitted to the data, and an 82% accuracy threshold. The two set sizes below this intersection were defined as ‘very easy’ and ‘easy’, and the two levels above it were defined as ‘medium’ and ‘hard’. If the intersection between the curve and the threshold was lower than four, participants were considered poor performers and excluded from the study (*n* = 11). While all four difficulty levels were used for the subsequent imaging visit, only the ‘easy’ and ‘hard’ levels were used during the TMS visits.

### 2.4. Targeting Approach

During the second visit, subjects participated in an MRI scanning (General Electric MRI scanner, B_0_ field strength = 3 Tesla) during which structural-T1-weighted (echo-planar sequence: voxel size = 1 mm^3^, TR = 7.148 ms, TE = 2.704 ms, flip angle = 12°, FOV = 256 mm^2^, bandwidth = 127.8 Hz/Pixel), T2-weighted (echo-planar sequence with fat saturation: voxel size = 0.9375 × 0.9375 × 2.0 mm^3^, TR = 4 s, TE = 77.23 ms, flip angle = 111°, FOV = 240 mm^2^, bandwidth = 129.1 Hz/Pixel), and diffusion-weighted scans (single-shot echo-planar: voxel size = 2 mm^3^, TR = 17 s, TE = 91.4 ms, flip angle = 90°, FOV = 256 mm^2^, bandwidth = 127.8 Hz/Pixel, matrix size = 128^2^, B-value = 2000 s/mm^2^, diffusion directions = 26) were obtained. Functional acquisitions (EPI-sequence: voxel size = 3.4375 × 3.4375 × 3.99 mm^3^, TR = 2 s, TE = 25 ms, flip angle = 90°, FOV = 220 mm^2^, bandwidth = 127.7 Hz/Pixel) were also acquired as participants performed the DRAT in the scanner. After preprocessing the images, functional data were analyzed using a general linear model (GLM) in which trial events were convolved with a double-gamma hemodynamic response function. The GLM examined BOLD responses during trials where the correct response was chosen in the behavioral task. Separate events were modeled for the array presentation (3-s duration), the delay period (5-s duration), and the response period (trial-specific RT duration). All incorrect and non-response trials were modeled identically, but separately, and were not considered in the results. 

The stimulation target was individually defined as the peak activation within the left lateral parietal cortex associated with a parametric increase in task difficulty during the delay period of the DRAT. According to the results obtained in our previous study [27], both set size (the number of letters in an array) and sorting steps (the minimum number of operations required to alphabetize the array) contributed to the difficulty of an individual trial. Therefore, to obtain a more accurate representation of increases in DRAT difficulty, a parametric delay-period regressor, defined by the interaction between set size and sorting steps, was used to estimate task difficulty. This parametric regressor was orthogonalized to the non-parametric delay-period regressor. At the first level, functional data were analyzed as individual runs. Second-level analyses combined data across runs for each subject using a fixed-effects model. This processing allowed for the definition of the stimulation target on individualized statistical maps that predicted the parametric increase in BOLD activity associated with increasing task difficulty if the peak activation reached a *z*-statistic value >2; or alternatively on the nonparametric delay-period map, if the peak did not reach this significance threshold. 

To constrain the stimulation target within the left lateral parietal cortex, a mask obtained from the group activation of 22 older adults who participated in our previous study was used [24]. The mask was defined as the overlap between the parametric interaction between set size and sorting steps (at *z* > 1) and the non-parametric delay period activity (at *z* > 1), therefore reflecting cortical regions that were generally activated by the task, but also specific to difficulty increase. The individual activation was then transformed back into subject space, and the peak activation within this mask was selected as the TMS target in the neuronavigation system (BrainSight, Rogue Research, Canada). 

To define the coil orientation, the coordinates from the TMS target were projected onto the scalp surface using a nearest neighbor approach and then projected slightly outwards to account for the subject’s hair thickness (Appendix A). Hair thickness was measured on each participant during the screening visit, using a depth gauge (Digital Tread Depth Gauge, Audew, Hong Kong; resolution 0.01 mm) installed on a custom-made plastic base placed over the center of the group parietal mask (Appendix A). The TMS coil was then oriented around the scalp normal vector so that the direction of the second phase of the induced E-field coincided with the inward-pointing normal vector on the sulcal wall closest to the brain target location. This pulse direction induced the strongest E-field and activation at the target [31,32]. The sulcal wall was identified using Freesurfer’s gyral/sulcal cortex classification ([33]: file lh.sulc), a byproduct of SimNIBS’ mri2mesh script during the brain surface extraction) and a brain surface point was chosen at the transition location in-between local concavity and convexity (|local curvature threshold| < 0.05) defining the sulcal wall. In order to compute the normal vector of that sulcal wall point, the surface normal of the triangles were averaged. The intended coil orientation was then entered in the neuronavigation system using the ‘twist’ tool. 

### 2.5. Stimulation Amplitude Approach

Rather than defining rTMS pulse amplitude according to a percentage of the motor threshold, as is frequently done in the literature, amplitude here was defined according to target-specific E-field values. While the motor threshold provides individualized information regarding the cortical reactivity of the motor cortex, it does not take into account differences in head anatomy and brain physiology between the motor cortex and other cortical regions within an individual. As such, traditional amplitude calibration based on the motor threshold may lead to substantial variation in the desired E-field strength in the targeted brain region. This may lead to response variability since the E-field strength is the key determinant of neural activation by TMS [34]. Therefore, in the present study, we selected the TMS pulse amplitude (coil current rate of change, di/dt) to induce a specific E-field magnitude, Eref, in the left lateral parietal region of interest (ROI) across subjects. 

To define Eref, computer simulations were used to estimate the E-field distribution within the parietal ROI induced when TMS was applied at an amplitude equal to the resting motor threshold in each of 9 subjects from a previous study [24] (see Appendix A). For each of the 9 subjects, a parietal ROI was constructed by individual fMRI activity (|z| > 0; within a group activity mask) registered to the individual’s space (FSL flirt [35]) within voxels classified as gray matter (SimNIBS: gm_only.nii.gz). The selected voxels were ranked according to their E-field strength, and a metric, E100, was defined as the minimum strength across the 100 voxels with the strongest E-field (Appendix A). The meaning of this metric is that 100 voxels in the ROI, corresponding to a volume of 100 mm^3^, have E-field strength larger than E100. The average E100 across the 9 subjects was calculated to be 56 V/m, which we set as our desired target E-field strength, Eref = 56 V/m. 

To select the individual TMS pulse amplitude in this study, computer simulations were performed to estimate the individual E-field distribution (analyzed for the left parietal ROI) and determine a TMS coil di/dt for which E100 = Eref in the ROI for each subject. Since TMS pulse di/dt scales linearly with the induced E-field, TMS was simulated for di/dt = 106 A/s, and a scaling factor was computed for di/dt to reach Eref for the hair thickness measured during the screening visit. The individual’s di/dt-value was determined for different hair thicknesses (in steps of 0.5 mm from scalp surface) and stored in a reference table (Appendix A). During the first TMS visit, the hair thickness at the exact stimulation location was re-measured and rounded to match the closest value in the table. The corresponding computed di/dt value was selected. The TMS amplitude, expressed as a percentage of the maximum stimulator output (MSO), was adjusted for the chosen di/dt value. The amplitude, together with the determined location and orientation (described in Section 2.4), were then experimentally applied. Two E-field strengths in the targeted region were experimentally tested, with E100 metric equal to either 80% Eref or 100% Eref. Resting motor threshold assessed during the screening visit was used to ensure that all stimulation intensities were below 130% of the resting motor threshold, and therefore within the published safety guidelines [36]. 

The computer simulations of the TMS-induced E-field were performed using the SimNIBS software package (Version 2.0.1; [32]). A computational model of each participant’s head was first generated employing co-registered T1- and T2-weighted MRI data sets to model major head tissues (scalp, skull, cerebrospinal fluid, gray and white brain matter) represented as tetrahedral mesh elements. Each mesh element was assigned a conductivity value based on its tissue association. The scalp, skull, and cerebrospinal fluid conductivities were set to isotropic values of 0.465, 0.010, and 1.654 S/m, respectively. The gray and white matter compartments were assigned anisotropic conductivities to account for the fibered tissue structures. This was accomplished within SimNIBS by co-registering diffusion-weighted imaging data (available for 14 out of 15 participants) and employing a volume normalization approach [37] which kept the geometric mean of the conductivity tensor eigenvalues equal to the default literature-based isotropic values of 0.275 and 0.126 S/m for gray and white matter, respectively. For the subject with missing DTI information, the latter values were assigned as isotropic conductivities. (Appendix A for individual subjects’ information)

### 2.6. TMS Procedure

During visits 3 to 6, participants performed the DRAT while active or sham rTMS was delivered to the individualized left lateral parietal target using an A/P Cool-B65 coil (MagVenture, Alpharetta, GA, USA). Twenty-five pulses of 5 Hz rTMS were delivered during the delay period of each trial (Figure 3). For every two trials with stimulation, one trial without stimulation was performed. This approach, successfully used in multiple studies by Luber et al. [9,38,39], theoretically allows time for neural activity in the stimulated region to return to its homeostatic baseline, allowing for greater range for the production of rTMS-induced plasticity and thus, potentially, greater rTMS effect on behavioral performance. The non-stimulated trials were excluded from subsequent analyses. The two intensities of stimulation, 80% Eref and 100% Eref, were applied on different days. Sham stimulation was applied using the same coil in placebo mode, which produced clicking sounds and somatosensory sensation via electrical stimulation with scalp electrodes similar to the active mode, but without a significant E-field induced in the brain [40]. This type of sham stimulation allows participants to stay blinded during the course of the experiment. Neuronavigation (BrainSight, Rogue Research, Canada) and real-time robotic control (SmartMove, ANT, the Netherlands) were used to ensure that the optimal coil position was maintained throughout the stimulation sessions.

On each TMS visit, subjects performed the DRAT at their two individually titrated difficulty levels (‘easy’ and ‘hard’). Twelve blocks of the DRAT task were performed (30 trials per block): one block without stimulation (No-Stim1), followed by five blocks of active or sham stimulation, one block without stimulation (No-Stim2), and five more blocks with the sham or active stimulation. The first 5 rTMS blocks in the first visit were always active rTMS at 100% Eref to ensure that the subjects were able to tolerate this stimulation amplitude, with the later 5 rTMS blocks being sham stimulation at output setting corresponding to the 100% Eref condition. For the three other visits, the intensities of stimulation were alternated by day, and sham and active stimulation were applied on the same day in random order. Random allocation, enrollment, and assignment was made through a Matlab script that was administered by the clinical research specialists. No adverse events or pain due to the stimulation were reported by any of the subjects. As noted above, our central hypothesis was that older adults would show a benefit for WM accuracy on the DRAT due to online rTMS, but only during the most difficult condition.

### 2.7. Statistical Analyses

Analyses were performed using the general linear model module of Statistica (TIBCO Software Inc., Palo Alto, CA, USA), normality was tested using Kolmogorov–Smirnoff tests, and multiple comparisons corrections were performed using Bonferroni correction. All the results are expressed as mean ± standard error.

## 3. Results

### 3.1. fMRI Activation during the DRAT

Figure 4 displays whole-brain results describing the parametric increase in BOLD-related activity with increasing difficulty in the DRAT, in the group of 15 completers. While not the basis of our optimized targeting technique, these results nonetheless demonstrate evidence of the active engagement of the fronto-parietal network during the memory task. 

### 3.2. Performance without rTMS

Four individualized difficulty levels (‘very easy’, ‘easy’, ‘medium’, and ‘hard’) were defined according to participant’s performance on the staircase version of the DRAT performed during visit 1. At the ‘very easy’ level, the absolute number of letters to maintain and alphabetize was found to be: 3 (*n* = 11), 4 (*n* = 3), or 5 (*n* = 1). During the rTMS sessions on visits 3 through 6, only ‘easy’ and ‘hard’ difficulty levels were used. To ensure that these difficulty levels were properly defined, to test the differences between ‘valid’ and ‘invalid’ trials, and to assess learning across time, repeated measure ANOVA was performed with the following within-subject factors: visit (visit 3, visit 4, visit 5, and visit 6), difficulty (‘easy’ and ‘hard’) and task condition (‘valid’ and ‘invalid’ trials). To prevent contamination of the data due to potential rTMS carryover effects, only behavior obtained during the first block without stimulation (No-Stim1) was considered. Trials for which the subjects did not answer (1.79 ± 0.57%) were excluded. A significant main effect of difficulty was found (*F*(1,13) = 112.55, *p* < 0.001) with higher accuracy for ‘E = easy’ (88.86 ± 3.32%) than for ‘hard’ (63.43 ± 3.25%) difficulty levels. No significant main effect of visit *F*(3,39) = 2.17, *p* = 0.11), task condition (*F*(1,13) < 1) or interaction between these factors were found. As such, it can be inferred that the difficulty levels were well defined by the staircase procedure. Subjects did not exhibit significant learning across visits, and accuracy was equivalent in the ‘valid’ and the ‘invalid’ trials.

### 3.3. TMS Dosing Results

As described in the methods section, TMS targeting used individualized coil position, coil orientation, and stimulation amplitude based on fMRI, structural MRI, and E-field data. The following sections will, therefore, present the individual and group results for each of these parameters.

#### 3.3.1. TMS Coil Position and Orientation

TMS coil position was constrained by a group mask defined on the left lateral parietal cortex (MNI coordinates at center of mask: −41; −64; 42, Figure 5 left). For each subject, the peak fMRI activation associated with difficulty increases during the DRAT within this mask was used to define the coil position. TMS coil orientation was then selected, such that the second phase of the induced E-field was perpendicular and directed into the nearest sulcal wall for each individual subject. The right panel of Figure 5 illustrates the final coil position and orientation for each subject.

#### 3.3.2. Stimulation Amplitude

The relationship between the individual TMS pulse amplitude inducing 100% E_ref_ at the parietal target and the resting motor threshold (Appendix A) was positive but not significant (r = 0.38, *p* = 0.16). The positive slope of the regression line likely reflects the fact that the motor threshold captures some anatomical factors such as the scalp-to-cortex distance that are correlated, to some degree, between various brain regions—this is the premise of conventional motor threshold-based dosing. While this correlation could be underpowered, given the small sample size, the lack of significance in the correlation appears to support the premise of our dosing approach, specifically that matching the E-field exposure of the target across subjects results in pulse intensities that are not predicted by motor cortex reactivity. The pulse amplitude based on E-field modeling was below resting motor threshold for some subjects (*n* = 8), while it was above this threshold for the others (*n* = 7). This inter-subject variability suggests that studies that dose based on a percentage of resting motor threshold may result in considerable within-group variability in the actual E-field that is induced in the cortex. In the current study, this potentially unwanted variability was controlled through appropriate dosing that considered the magnitude of the E-field in the desired cortical target. It should be noted that the resultant individual pulse intensities did not exceed the rTMS safety guidelines of <130% of resting motor threshold for trains of less than 10 s in any of the subjects [36].

### 3.4. rTMS Effects

#### 3.4.1. Cumulative rTMS Effects

In the current design, active and sham stimulation were performed on the same day. To ensure that no cumulative carryover effects contaminated the effect of one type of stimulation over the other, the block of trials performed either immediately before (No-Stim1) or immediately after each type of stimulation (No-Stim2_AfterActive, and No-Stim2_AfterSham) were analyzed separately. Trials for which subjects did not respond were excluded (1.70 ± 0.49%). One way ANOVA, performed on accuracy across these three blocks, did not reveal significant differences between blocks performed before stimulation (No-Stim1 = 76.38 ± 2.36%), blocks performed after active rTMS (No-Stim2_AfterActive = 76.29 ± 2.05%), or blocks performed after sham rTMS (No-Stim2_AfterSham = 78.08 ± 2.32, *F*(2,28) < 1). This result suggests that no carryover effects persisted, and thus subsequent rTMS effects were not contaminated by the former type of stimulation.

#### 3.4.2. Omnibus rTMS Effects on the DRAT

To assess rTMS effects on performance during the DRAT, repeated-measures ANOVA was run on accuracy scores with the following within-participant factors: task condition (‘valid’ and ‘invalid’), stimulation type (active and sham), stimulation amplitude (80% and 100% E_ref_), and Task Difficulty (Easy and Hard). Trials for which the subjects did not answer (1.93 ± 0.4%), and trials during which stimulation was not applied (33.3% No-Stim trials), were excluded from this analysis. 

This ANOVA produced non-significant main effects of task condition (‘valid’: 72.84 ± 6.53%, ‘invalid’: 71.24 ± 5.86%, *F*(1,14) < 1) and stimulation type (active: 71.70 ± 6.29%, sham: 72.38 ± 6.12%, *F*(1,14) < 1) on task accuracy. A significant main effect of stimulation amplitude was observed, however, with lower accuracy when stimulation was applied at 100% E_ref_ (70.19 ± 6.26%) compared to stimulation applied at 80% E_ref_ strength (73.89 ± 6.12%; *F*(1,14) = 10.15, *p* = 0.007, η^2^ = 0.42). A significant main effect of difficulty was also observed (*F*(1,14) = 117.4, *p* < 0.001, η^2^ = 0.89), with lower accuracy for hard trials (56.59 ± 5.60%) compared to easy trials (87.49 ± 3.6%). A significant three-way interaction was also found between task condition, stimulation type and stimulation amplitude (*F*(1,14) = 5.14, *p* = 0.04, η^2^ = 0.27). The decomposition of this interaction revealed that, only for the ‘invalid’ trials, active rTMS applied at 100% E_ref_ (69.14 ± 6.4%) tended to disrupt accuracy compared to active rTMS applied 80% E_ref_ (74.25 ± 6.1%) (*F*(1,14) = 4.19, *p* = 0.088). This suggests that applying rTMS at a stronger amplitude leads to a larger rTMS effect. ANOVA also revealed a significant interaction between stimulation type and difficulty (*F*(1,14) = 9.70, *p* = 0.008, η^2^ = 0.41). Bonferroni corrected post-hoc comparisons revealed that while no differences were found between active (87.97 ± 3.7%) and sham stimulation (87.00 ± 3.71%) for ‘easy’ trials (*F*(1,14) = 2.22, *p* = 1), for ‘hard’ trials active rTMS significantly decreased accuracy (55.42 ± 5.50%) compared to sham rTMS (57.76 ± 5.70%; *F*(1,14) = 4.12 *p* = 0.045; Table 2). This result indicates that rTMS only disrupts performance for harder memory trials.

#### 3.4.3. Follow-Up Tests on the Effects of Stimulation Amplitude

To further investigate differences between stimulation applied at 80% and 100% E_ref_, a separate follow up ANOVA was performed with the factors difficulty (‘easy’ and ‘hard’) and stimulation type (active and sham) for each stimulation amplitude. Results for 100% E_ref_ showed that accuracy was significantly decreased in ‘hard’ trials (60.02 ± 2.65%), relative to ‘easy’ trials (88.38 ± 2.65%; (*F*(1,14) = 148.3, *p* <0.001, η^2^ = 0.91). While the main effect of stimulation type was not significant (active: 73.37 ± 4.72% vs. sham: 75.04 ± 4.29%, *F*(1,14) = 2.89, *p* = 0.11, η^2^ = 0.17), there was a significant interaction between stimulation type and difficulty (*F*(1,14) = 6.42, *p* = 0.02). Post-hoc Bonferroni comparisons showed that, while no differences were found between active and sham stimulation at the ‘easy’ difficulty level (active: 88.49 ± 2.79% vs. sham: 88.28 ± 2.61%, *p* = 1.00), active rTMS did significantly decrease accuracy (58.26 ± 2.39%) compared to sham rTMS (61.79 ± 2.40%; (*F*(1,14) = 6.68, *p* = 0.03) on the ‘hard’ trials. 

For the ANOVA performed with rTMS applied at 80% E_ref_, a main effect of difficulty was found with lower accuracy for ‘hard’ trials (63.54 ± 2.28%) compared to the ‘easy’ ones (90.84 ± 2.24%, *F*(1,14) = 136.0, *p* < 0.01, η^2^ = 0.91). No differences were found for stimulation type (*F*(1,14) < 1). Interestingly, the interaction between difficulty and stimulation type was not significant for this stimulation amplitude (*F*(1,14) = 2.02, *p* = 0.17). These results, therefore, show that the disruptive effects of rTMS for harder memory trials are specific to the highest stimulation amplitude. However, this effect needs to be interpreted with caution because, as indicated by the non-significant interaction between E-field strength, stimulation type, and difficulty (*F*(1,14) < 1), in the larger ANOVA, it did not reach omnibus significance.

At the individual level, when computing the rTMS effect as a percentage of change between active and sham rTMS the results showed that when stimulation was applied at 100% E_ref_, 9 out of 15 subjects showed performance disruption, while 7 showed disruption when rTMS was applied at 80% E_ref_ (Figure 6). Interestingly, 4 out of 7 subjects (2, 6, 12, and 13) who experienced a disruptive effect at 80% Eref, exhibited a facilitation effect at 100% Eref, potentially suggesting an individual and non-linear effect of rTMS intensity on behavioral outcomes.

### 3.5. Effect Sizes Comparison with Prior Study

As mentioned in the introduction, the goal of the current study was to optimize the facilitatory effects of rTMS observed in [24], where 5Hz rTMS was applied over the left dorsolateral prefrontal cortex. Here, we wished to maximize the difference between active and sham rTMS by stimulating the parietal cortex, which has been shown to be more activated during the DRAT [27], and by modifying some stimulation parameters. To compare the effect sizes for these two studies, Cohen’s d was calculated from the individual conditions that yielded the largest effects in each study (hardest difficulty level in the ‘invalid’ trials in [24] and 100% E_ref_ strength for the current study). Cohen’s d was calculated as follows: $$d=\frac{m_{1}-m_{2}}{\sqrt{\frac{s_{1}{}^{2}+s_{2}{}^{2}}{2}}}$$
where *m*_1_ and *m*_2_, and *s*_1_ and *s*_2_ are the mean and standard deviation of accuracy for active and sham, respectively. 

Results show a modest rTMS-induced effect size for both studies and highlight the opposite directionality of this effect. While the effect size was negative (Cohen’s d = −0.38) in [24], it is positive in the current study (Cohen’s d = 0.34). This contrast illustrates that, instead of optimizing the facilitatory rTMS effect obtained in the former study, modifications made in the current study reversed the rTMS effect, while producing relatively equivalent effect sizes. 

## 4. Discussion

This study was conducted to test whether parameter-optimized rTMS, delivered to the left lateral parietal cortex, could enhance working memory manipulation in healthy elderly subjects. In our previous study [24], applying 5Hz rTMS over the left prefrontal cortex increased young and elderly participants’ accuracy, but the effect was small. Therefore, in the current study, the goal was to modify the stimulation target and optimize stimulation parameters in order to produce larger performance enhancements. However, these changes led to an opposite pattern of findings wherein active stimulation yielded a small performance impairment relative to sham stimulation, with a similar effect size. The following sections discuss these patterns of effects, as well as the innovative targeting and dosing approaches used in this study.

### 4.1. Site- and Timing-Specific rTMS Effects on WM Manipulation

In our studies, online rTMS was applied with a goal of enhancing the manipulation of information in WM, as assessed by performance on the DRAT. As reported in [27], group analysis of fMRI acquired during the second visit of these studies revealed that when participants are mentally maintaining and alphabetizing letters during the delay period of the DRAT, the left lateral parietal cortex produced strong activation. This finding is consistent with the role of the parietal cortex in symbolic computations [41,42,43] and led to targeting of the parietal cortex, rather than DLPFC, in the current study. As such, contrasting findings between the two cohorts reflect the site-specificity of rTMS effects on WM.

While it is widely reported in the literature that online rTMS applied during a task induces a temporary ‘virtual lesion’, evidence of performance enhancement has also been found ([7], for a review). One factor likely mediating the opposite effects of online rTMS is the modulation of endogenous task-related oscillatory dynamics in a manner that is specific to the timing of ongoing processes [10]. Working memory tasks, in particular, have been associated with 5 Hz theta-band coupling between frontal and parietal regions during the memory retention period, which increases parametrically with memory load [44]. However, contrary to such expectations, applying 5 Hz rTMS during the delay period disrupted participants’ accuracy and suggested a more complex interaction between site-specific and timing-specific rTMS effects. One possible explanation for this discrepancy is that while rTMS may engender certain oscillatory patterns at the site of stimulation, there exists some general latency after the end of the last TMS pulse for that entrainment to emerge. Thus, rTMS applied to a region during the time it is engaging in task-essential processing will disrupt performance, while stimulation trains of the appropriate frequency applied prior to that processing can enhance performance, possibly due to entrainment of functional oscillations prior to the essential cortical activity [45,46]. The fMRI evidence suggests the essential processing in parietal cortex needed for the DRAT occurred during the delay period, with rTMS applied in that period injecting random noise that disturbed performance, while delay period 5 Hz rTMS to DLPFC may result in pre-processing activity, possibly through entrainment of theta frequency, that may enhance processing there during the subsequent probe period. As such, applying rTMS over the parietal cortex, before the encoding might lead to performance enhancement; however, more investigation is needed to determine the stimulation timing parameters that will lead to such enhancement. 

### 4.2. Moderate Effect Sizes

In both of our studies, rTMS effects were found to be small and limited to the most difficult conditions of the task. Interestingly, several studies report that non-invasive brain stimulation effects on behavioral performance can be difficulty-specific [9,13,38], and also subject-specific, with baseline performance affecting the subsequent brain stimulation effect. For example, participants with a lower performance at baseline are the ones benefitting the most from stimulation (e.g., [14,47,48]). In our studies, the difficulty levels (i.e., the number of letters to mentally manipulate) were individually-titrated based upon subjects’ performance during their first visit. They were defined according to the intersection between a sigmoid curve, fitted to the data, and an 82% accuracy threshold, with participants excluded if the intersection between the curve and this threshold was at set sizes lower than 4. While this criterion was added to ensure that all participants were able to properly perform the task over a range of difficulty that began above a floor performance level, it could also constitute a limitation in our study, and, if poorer performers could benefit most from TMS with our task, it might explain the limited effect sizes observed here. Future studies may wish to include a larger range of behavioral performance Given the importance of brain oscillations [49], initial brain state [50], and baseline performance in brain stimulation outcomes, a potential way to improve effect sizes would be to tailor the pattern of brain stimulation according to each individual’s specific neuronal activity, in an open- or closed-loop dependent manner as reviewed by Thut et al. [51].

### 4.3. Electric-Field-Based TMS Dosing 

We adopted a novel individualized TMS targeting strategy that combined fMRI, structural MRI, and E-field modeling to select the TMS coil position, orientation, and pulse amplitude. The TMS coil was centered over the individualized peak fMRI activation, within a group mask based on previously reported fMRI data for this task [24], and associated with difficulty level in the DRAT. The subject’s brain anatomy, imaged via MRI, was then used to define the optimal coil orientation such that the E-field pulse was perpendicular and directed into the nearest sulcal wall. As such, this approach allowed us to address the potential confounds associated with cortical thinning associated with older adult samples and builds on previous research in the motor cortex showing the lowest threshold for motor evoked potentials when the induced E-field is perpendicular and flowing into the sulcal wall [32,52,53]. This observation has been replicated in various other cortical regions [31,32,54]. The likely explanation for the optimality of current perpendicular to the sulcal wall is that this current orientation induces the strongest E-field in the corresponding gyrus [31]. The likely reason for the sensitivity to the current direction (current flowing into the gyrus) is the morphology and orientation of pyramidal neurons in the cortex [34]. Furthermore, we found no effect of cortical thickness across subjects on the observed TMS-related effects on performance, suggesting that E-field modeling is a good method to prevent this bias (Appendix A). Since these observations appear to be generalizable to any area of the cerebral cortex, we adopted this approach in our study as a means to minimize the pulse amplitude required for stimulating the target and to maximize the stimulation focality. 

The traditional approach for individualizing the amplitude of TMS pulse is to scale the coil current based on the easily-observable resting motor threshold response. While this approach is appropriate for stimulation of the primary motor cortex, where it captures the underlying anatomical and physiological variability across subjects, its use in other brain regions has significant limitations. First, the individual location of the muscle representation in the primary motor cortex affects the motor threshold but is unlikely to match the location of the non-motor target area on its respective gyrus. Second, the local anatomy of the head in the vicinity of the target can vary individually. Third, the local physiology may differ from that of the motor cortex in an individual manner. Therefore, we opted to match the delivered E-field strength at the cortical target across subjects. The reference E-field strength, E_ref_, was selected based on estimates of the average stimulation delivered in prior studies with a similar target. We introduced a metric, E_100_, that ensures that a fixed volume of the cortical target is exposed to an E-field strength at or above the reference value, while also reducing the sensitivity to computational outliers that may impact other E-field metrics such as peak or mean value. While the design of the current study did not compare the effect of rTMS applied with dosing based on the motor threshold versus the simulated E-field strength, the results highlighted a significant difference between stimulation applied at 80% or 100% of E_ref_. Indeed, stronger disruptive rTMS effects were associated with stimulation applied at higher amplitudes. This suggests that the E-field-based dosing approach produced sufficiently consistent rTMS effects across subjects to differentiate the two amplitude conditions. More investigation is needed to explore the value of this new dosing approach.

## 5. Conclusions

5 Hz rTMS to the left lateral parietal cortex during the delay period of the DRAT was found to impair WM behavior of healthy older adults. When considered in light of Beynel et al. [24], the effects can be interpreted as site- and timing-specific effects of rTMS on WM manipulation. Collectively, across these two cohorts, findings demonstrate the ability of online rTMS to up-regulate and down-regulate WM by stimulating prefrontal and lateral parietal cortices, respectively. This information is important for further clarifying how rTMS may be used to therapeutically treat disorders where memory is impacted. Future investigation is warranted to refine the parameters under which these effects are present, and to explore how targeting and dosing considerations can be adjusted to optimize rTMS efficacy. 

## Figures and Tables

**Figure 1 brainsci-10-00255-f001:**
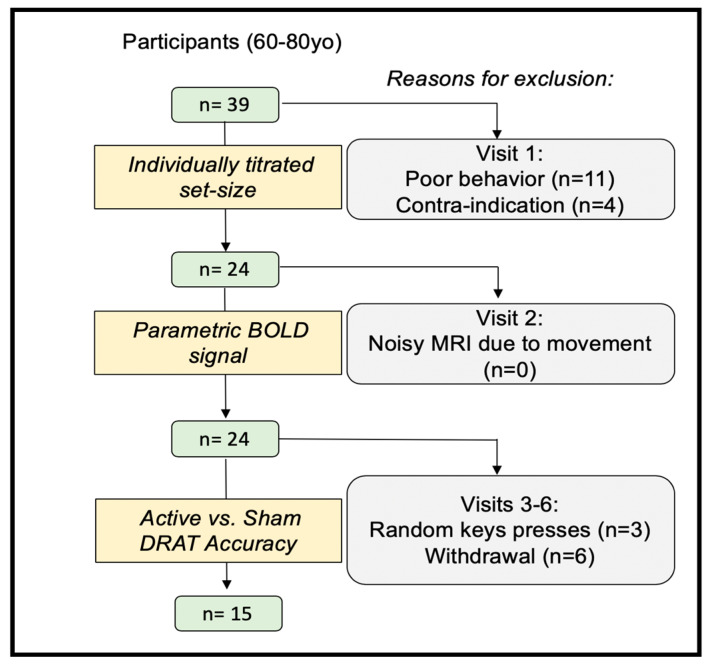
Consort diagram showing the recruitment, exclusion, and inclusion numbers (DRAT: Delayed-Response Alphabetization Task).

**Figure 2 brainsci-10-00255-f002:**
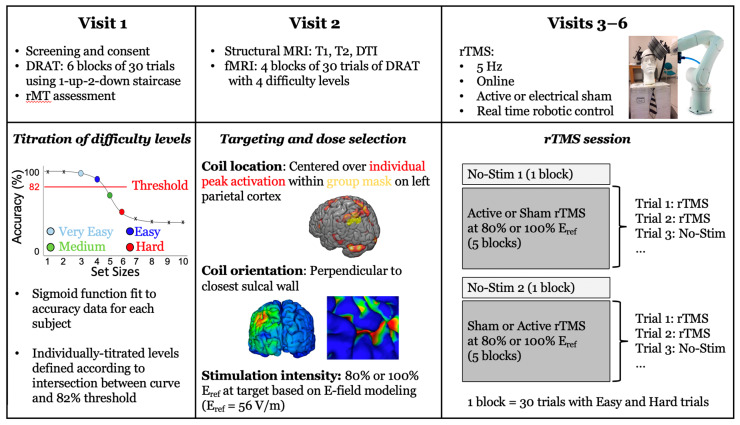
Illustration of the full experimental protocol across 6 visits. For visits 1 and 2, data processing procedures are illustrated in the bottom panel. For visits 3–6, the bottom panel illustrates the experimental procedure in greater detail (T1: T1-weighted images, T2: T2-weighted images, DTI: diffusion tensor imaging, rTMS: repetitive transcranial magnetic stimulation).

**Figure 3 brainsci-10-00255-f003:**
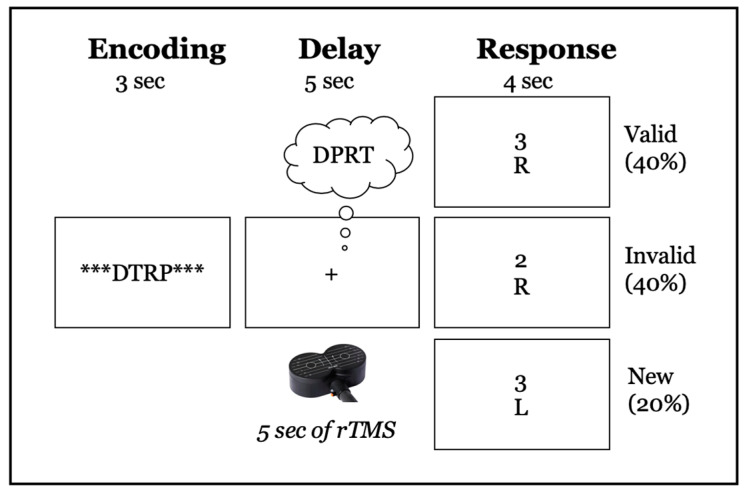
Schematic illustration of DRAT task. The top row shows an array of 4 letters (DTRP), presented for 3-s, that participants have to encode. Row two shows the 5-s delay period, during which participants had to maintain and reorganize the letters into alphabetical order (DPRT in the figure). Row three shows examples of the three possible responses: ‘new’: the letter was not in the original array; ‘valid’: the letter was in the array, and the number represented the correct position in the alphabetical order; ‘invalid’: the letter was in the array, but the number did not match the correct serial position when alphabetized.

**Figure 4 brainsci-10-00255-f004:**
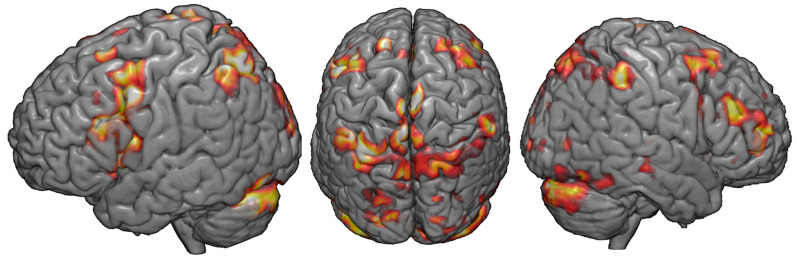
fMRI response to the DRAT task: Whole-brain results describing the parametric increase in BOLD-related activity with increasing task difficulty (2 < *z* < 3).

**Figure 5 brainsci-10-00255-f005:**
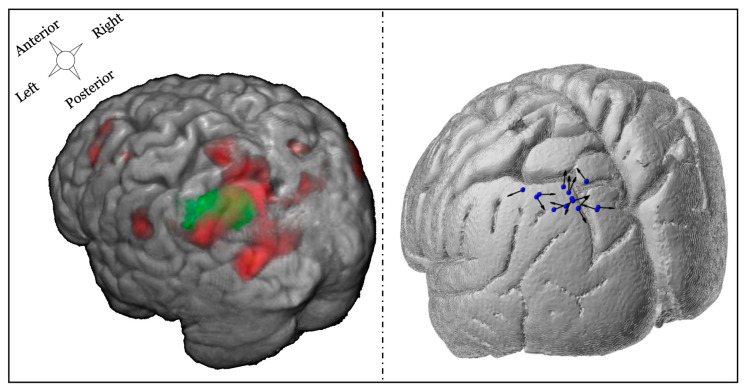
Illustration of fMRI targeting and final TMS positioning. Left panel: example of one individual subject combining the group mask over the left parietal cortex (green) combined with individual fMRI activation (red) associated with the parametric increase in difficulty during the delay. Right panel: TMS coil position and orientation for each participant. The blue spheres represent the coil location, and the black arrows correspond to the direction of the second phase of the induced E-field pulse (some of the arrowheads are not visible because of the 3D view).

**Figure 6 brainsci-10-00255-f006:**
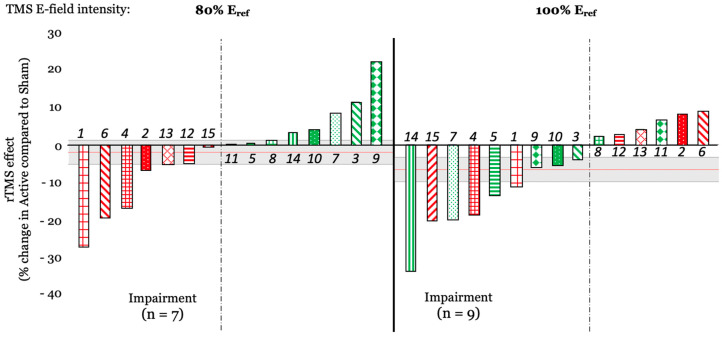
rTMS effect (expressed as a percentage of change between active and sham rTMS) for rTMS applied at 80% E_ref_ (left) and at 100% E_ref_ (right). Each bar is color-coded for each subject and based on rTMS effect at 80% Eref; numbers in italics represent the subject number. The red horizontal line shows the averaged rTMS effect across each subject (mean of −2.03%, and −6.66% for 80% E_ref_ and 100% E_ref_, respectively). The gray rectangle represents the standard error (SE = 3.19% and 3.26% for 80% E_ref_ and 100% E_ref_, respectively).

**Table 1 brainsci-10-00255-t001:** Baseline demographic for included participants.

**Number of completers**	15
**Age** (Mean ± SD)	66.13 ± 5.50 years old
**Gender**:	
Number of Females	4
Number of Males	11
**Number of education years** (Mean ± SD)	17.33 ± 1.79 years

**Table 2 brainsci-10-00255-t002:** Average accuracy for the difficulty levels and the stimulation type.

	Easy	Hard
**Active rTMS**	87.97 ± 3.70%	55.42 ± 5.50%
**Sham rTMS**	87.00 ± 3.71%	57.76 ± 5.70%
***p*-values**	1.00	0.045

## Data Availability

https://osf.io/ausq9/.

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
