# Peer review of "Site-Specific Effects of Online rTMS during a Working Memory Task in Healthy Older Adults"

_brainsci, 2020, doi:10.3390/brainsci10050255_

Round 1

Reviewer 1 Report

In “Site-Specific Effects of Online rTMS during a Working Memory Task in Healthy Older Adults” Beynel and colleagues investigated the impact of online theta rhythmic TMS (applied over an fMRI-defined left parietal cortex) on a delayed-response alphabetization task. They varied task difficulty (easy, difficult trials – define by a staircase procedure for each participant) and TMS intensity (using an interesting and new approach) to show that for difficult trials and high E-field TMS intensity, participants’ performance tends to decrease with stimulation as compared to sham.

The study is well written. However, in its current form, it looks more like a methodological articel describing an interesting new approach for determining TMS targeting and intensity than a cognitive neuroscience paper on Working memory (WM).

I have some major comments than should be considered before publication.

Major comments:

  • The recent literature on theta-rhythmic TMS during working memory is surprisingly totally ignored in the current version: Indeed several studies have shown that enhancing theta-band activity and synchrony in the fronto-parietal network can enhance participants’ performance during WM. Theta rhythmic TMS studies during WM such as Albouy et al. 2017 (Neuron), Violante et al. 2017 (eLife), Hoy et al. 2015 (Cereb Cortex) should be cited and discussed as well as tACS studies (e.g. Alekseichuk et al., 2016, Current Biology also during WM). I also strongly recommend to cite  and integrate recent reviews on the rhythmic TMS topic such as Hanslmayr, Axmacher, Inman, 2019 (Trends Neurosci. WM), Albouy et al. 2018 (Ann NY Acad Sci., WM), and the more general review by Polanía, Nitsche, Ruff  2018 (Nature Neuroscience). These articles should be cited and discussed in line with the present study.
  • Along these lines, the small effect (behavioral change of about 2.3%, d = 0.34) observed in the present study should be discussed according to this literature (previous articles mainly tested young adults, compare stimulus intensity, inclusion of bad performers). A more detailed discussion describing authors’ hypotheses for explaining the discrepant results should be added. In the current version, the discussion focuses more on the methodology. That’s interesting, but the authors should consider changing the topic of the paper to a methodological paper.
  • For the fMRI data, the first level maps were masked using a mask from a previous fMRI study (to define the TMS target for each participant). The rational is unclear. Please develop. How about using a mask from the current group of participants ? (not independent form the data but the rational here is to define the optimal target region for this group of participants).
  • The fMRI results should be presented in more details: it is important for the reader to evaluate if the task recruits the classic fronto-parietal WM network and if the BOLD activity in this network (not only on the parietal peak as described in the paper) varies as a function of task difficulty. Indeed the current version, the authors only report that “The stimulation target was individually defined as the peak activation within the left lateral parietal cortex associated with a parametric increase in task difficulty during the delay period of the DRAT.” This makes sense but it would be very interesting to see if this peak is connected to the fronto-parietal network.
  • The objective of the study is to enhance behavioral performance with non-invasive brain stimulation: it is thus very surprising to see that poor performers (who have, by definition, more room for improvement) were not included in the TMS protocol. It has been reported that theta rhythmic TMS effects on WM performance are stronger for participant showing poor performance without stimulation (Albouy et al. 2017). This fits well with the current observation (and the previous article from the authors) that the TMS effects are observed only for difficult trials. The exclusion of the bad performers should be more justified and the authors should consider this as a limitation of the study (e.g. poor performers might have shown an effect on easy trials)
  • Figure 5 should be moved to supplementary material, except if the authors change the topic of the paper to a methods article

Reviewer 2 Report

This is an excellent, rigorous paper that used a pre-registered, randomized, single-blind, placebo-controlled design to test the hypothesis that online 5 Hz repetitive TMS to left lateral parietal cortex during working memory manipulation would enhance performance in elderly adults. Contrary to the hypothesis, the authors found a performance decrement after active TMS compared to sham, particularly when the working memory task was more difficult (larger set size of letters to alphabetize). Despite being opposite in directionality, this effect was of a similar size to the authors’ previous findings, in which a performance benefit was observed with rTMS to lateral prefrontal cortex. These conflicting findings were therefore taken as evidence for site- and timing-specific effects of rTMS on working memory. In addition to being conceptually important, this paper provides a fine example of a TMS study done well. In addition to pre-registration, great care was taken to individualize behavioral thresholds, as well as TMS targets (and coil orientation) and TMS intensities on the basis of electric field modeling.

I have only a few minor comments:

  • About the E-field modeling generally: Perhaps this has already been done, but if you use the same procedure to choose stimulation intensities over motor cortex, how close do you end up to the measured motor threshold?
  • Page 3, lines 118-120: This description of the methods used to determine the sample size is unclear and could use elaboration. How was the expected effect size determined?
  • Page 10, lines 348-350: While I believe that your approach is an improvement over choosing the stimulation intensity based on motor thresholding, I’m not sure that you can lean too hard into the lack of significance of the correlation here – a 15-person correlation is certainly under-powered.
  • Figure 6: It would be useful to know how much correspondence there is between these two plots. Are participants reliably affected in the same direction at both intensities? It may look too busy, but something like assigning each participant a unique color might be helpful.
  • Figure S3: These histograms look clipped on the y-axis

Even more minor:

  • “E-field” is used in the introduction before being spelled out
  • Table 2: the actual p-value would be more helpful than “NS”
  • Page 9 line 331: I think this should say “increases”

Round 2

Reviewer 1 Report

I thank the authors for considering all my comments and I recommend publication.